# Administering Nitric Oxide (NO) with High Flow Nasal Cannulas: A Simple Method

**Vladimir L. Cousin** [1,2,*], **Raphael Joye** [3] **and Angelo Polito** [1]

1 Pediatric Intensive Care Unit, Department of Pediatrics, Gynecology and Obstetrics, University Hospitals and Faculty of Medicine, University of Geneva, CH-1211 Geneva, Switzerland

2 Pediatric and Neonatal Intensive Care Unit, Department of Pediatrics, Gynecology and Obstetrics, University Hospitals and Faculty of Medicine, Rue Willy Donzé 6, CH-1205 Geneva, Switzerland

3 Pediatric Cardiology Unit, Department of Pediatrics, Gynecology and Obstetrics, University Hospitals and Faculty of Medicine, University of Geneva, CH-1211 Geneva, Switzerland

* Correspondence: vladimir.cousin@hcuge.ch; Tel.: +41-22-37-24513; Fax: +41-79-55-25488

**Highlights:**

**What are the main findings?**

- A simple method to administer Nitric Oxide using the SoKINOX® (Air Liquide, Paris, France) Nitric Oxide delivery device with HFNC devices.

**What is the implication of the main finding?**

- It is suitable for both adult and pediatric circuits (Optiflow®, Fisher & Paykel Healthcare, Auckland, New Zealand).

**Abstract:** Inhaled nitric oxide (iNO) is a pulmonary vasodilator that plays an important clinical ICU role. The administration of iNO is usually performed through an endotracheal tube, but spontaneously breathing patients might also benefit from iNO administration. The use of the non-invasive administration of iNO through high-flow nasal cannula (HFNC) allows for NO delivery in spontaneously breathing patients who still need supplemental oxygen and positive airway pressure. A simple method to administer NO through HFNC is described here using standard commercially available NO administration and HFNC.

**Keywords:** nitric oxide; pediatric intensive care unit; pulmonary hypertension; high-flow nasal cannula





Inhaled nitric oxide (iNO) is a pulmonary vasodilator that plays an important clinical role in both adult and pediatric patients with pulmonary hypertension hospitalized in the intensive care unit [1]. Indeed, iNO decreases pulmonary arterial pressure and pulmonary vascular resistance without impacting systemic hemodynamic [1,2]. In addition, nitric oxide may have some antimicrobial and antiviral properties, making it a largely studied agent in viral pneumonia [3]. The administration of iNO is usually performed through an endotracheal tube. However, spontaneously breathing patients might also benefit from iNO administration. The use of the non-invasive administration of iNO through high-flow nasal cannula (HFNC) allows iNO delivery in spontaneously breathing patients who still need supplemental oxygen and positive airway pressure.

Some reports have described the use of iNO administered through non-invasive ventilation or HFNC for the treatment of pulmonary hypertension [4], postoperative management in cavo-pulmonary derivation [5], or hepatopulmonary syndrome [6,7]. In the recent COVID-19 pandemic, use of iNO trough HFNC was also studied to prevent further respiratory failure and intubation [8]. Nonetheless, iNO administration may be technically challenging, as specific devices are required to administer and monitor iNO delivery [9]. Despite the numerous different cylinder-based delivery systems largely used in intubated

patients, there is a lack of description regarding a practical method to deliver iNO without tracheal intubation using HFNC [9]. A simple method to administer NO through HFNC is described here, and this may allow easier access to iNO in non-intubated pediatric and adult patients for both clinical and research purposes.

SoKINOX® device (Air Liquide, Paris, France), with a NO administration/monitoring kit (Getinge, Solan, Sweden), was used for NO gas administration. We used HFNC devices that are suitable for adult as well as pediatric patients (Optiflow®, Fisher & Paykel Healthcare, Auckland, New Zealand). It is possible to use two types of montage for pediatric (Figure 1) and adult patients (Figure 2). The two montages share common features (i.e., the NO injection line connection to the humidifier). In both cases, the humidifier is used as a mixing chamber for inspiratory gases ($O_2$, air, and NO), and the sample line was placed as close as possible to the patient.

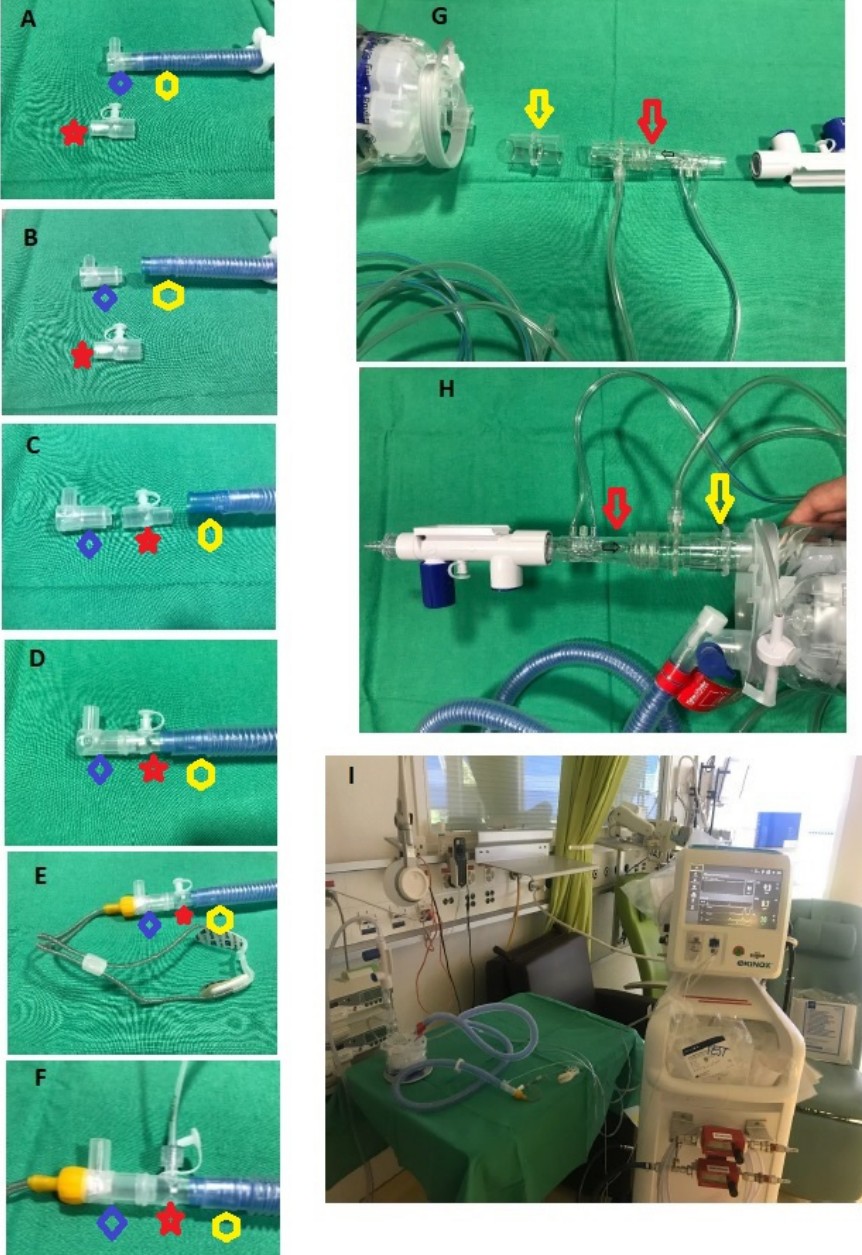

**Figure 1. Pediatric high flow nasal cannula montage for iNO administration. (A)** Luerlock adaptator (red star) (Fisher & Paykel Healthcare, Auckland, New Zealand) and the main flow pipe (yellow

hexagon) with the adaptor for the cannula (blue diamond); (**B**) separation of the tip of the main flow pipe; (**C**) interposition of the Luerlock between the adaptor of the cannula and the main flow pipe; (**D**) assembly of the 3 parts: cannula adaptor, Luerlock, main flow pipe; (**E**) final montage with high flow nasal cannula; (**F**) final montage with sampling line connected to the Luerlock; (**G**) NO injection line (red arrow) with the humidifier, a 22 mm tubing (yellow arrow), and the flow regulator; (**H**) final montage; (**I**) complete montage for pediatric HNPC (Optiflow® (Fisher & Paykel Healthcare, Auckland, New Zealand), using the SoKINOX® (Air Liquide, Paris, France). Red star: Luerlock adaptor; yellow hexagon: main flow pipe; blue diamond: adaptor for the cannula; red arrow: NO injection line; yellow arrow: 22 mm tubing.

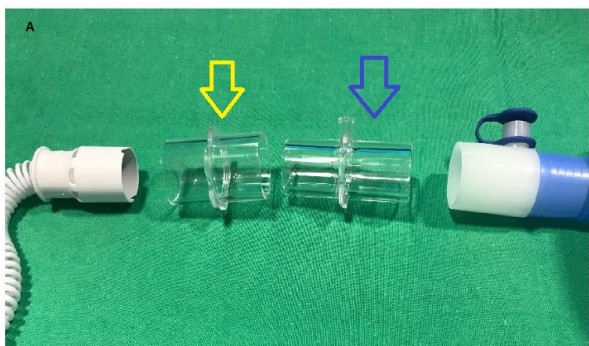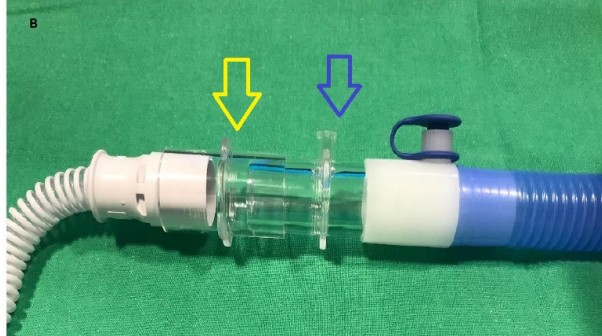

**Figure 2.** Adult high flow nasal cannula montage for iNO administration. (**A**) Connection of a T-piece from the NO administration/monitoring kit (blue arrow) (Getinge, Solan, Sweden) and use of a tubing of 22 mm from the kit (yellow arrow). (**B**) Final montage after assembling together the cannula, the 22 mm tube, the T-piece, and the main flow pipe. Yellow arrow: 22 mm tube. Blue arrow: T piece for NO monitoring line.

For the pediatric kit, the tip of the main flow pipe must be taken off in order to insert a Luerlock adaptor (Fisher & Paykel Healthcare, Auckland, New Zealand) (Figure 1A,B). Then, the adaptor for the cannulas is connected to the Luerlock, and the nasal canula itself can be connected to the main NO delivery system (Figure 1C–F). The NO injection line is connected to the humidifier using tubing furnished with the kit. A 22 mm tubing is placed between the humidifier and the injection line. For the adult kit, the NO injection line is connected to the humidifier using a single 22 mm tubing supplied by the manufacturer. As for the pediatric kit, the 22 mm tubing is then placed between the humidifier and the injection line, (Figure 1G,H). For the sample line, a sample-T-piece from the NO administration/monitoring kit (Getinge, Solan, Sweden) and a 22 mm tubing are needed to connect the cannula, the iNO sensor, and the main flow pipe (Figure 2A,B). The T-piece is connected to the main flow pipe, and then the 22 mm tube is used to connect the nasal cannula to the main NO delivery system.

We tested the montage on a life-like dummy patient with HFNC in place. This bench test confirmed the tightness of the montages and the accuracy of the sample line. We measure the NO concentration at the sample line at air flow of 6 L/min, 12 L/min, and 30 L/min. At each flow, the measured concentration quickly reached the desired concentration goal (10 ppm) and remained stable at such a concentration. We believe that the method of iNO administration described is effective, simple to use, and may be safely employed in spontaneously breathing patients needing HFNC treatment. It would allow for NO delivery while beneficiating from HFNC support, for which clinical indications have significantly increased in recent years in both adult and pediatric patients [10,11].

**Author Contributions:** V.L.C. contributed to the conception and design. V.L.C. and R.J. performed the bench tests and montage pictures. The first draft of the manuscript was written by V.L.C. V.L.C., R.J. and A.P. commented on previous versions of the manuscript, have read and agreed to the published version of the manuscript. All authors have read and agreed to the published version of the manuscript.

**Funding:** The authors declare that no funds, grants, or other support were received during the preparation of this manuscript.

**Institutional Review Board Statement:** Not applicable.

**Informed Consent Statement:** Not applicable.

**Data Availability Statement:** Not applicable.

**Conflicts of Interest:** The authors declare that they have no conflicts of interest to report.

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
