# Peer review of "Administering Nitric Oxide (NO) with High Flow Nasal Cannulas: A Simple Method"

_arm, doi:10.3390/arm92010012_

Round 1
Reviewer 1 Report
Comments and Suggestions for Authors
Cousin et al. report a novel non-invasive methodology for delivery of inhaled nitric oxide (iNO) through high-flow nasal cannulas (HFNCs) for potential therapeutic application in spontaneously breathing patients. iNO is of significant interest as a rescue measure for patients at risk of decompensation and endotracheal intubation. Critically, little work has described the combination of iNO with HFNCs. iNO administration has recently been described with HFNCs for ARDS in COVID-19 patients, but in the early stages of research conflicting results have often been found on the efficacy of the technology for improved clinical outcomes (DOI: 10.1177%2F11795484211047065). Cousin et al. report a novel device configuration strategy for achieving iNO therapy with configurations applicable to both adult and pediatric patients which may serve as a step forward in investigating the efficacy of the combination strategy in clinical outcomes. This manuscript is of high quality and suitable for publication in Advances in Respiratory Medicine following consideration of some minor revisions:
11. In the introduction, several examples of the benefits of iNO administration are listed for pulmonary and cardiovascular conditions. Authors should also tie in the recent interest in iNO for treatment of respiratory infections and further comment on the utility of the enclosed technology for such applications. Some prior work is described by: Biomedicines 2022 Feb 3;10(2):369 and Applied Materials Today 22(2021) 100887.
22. iNO treatment is stated as technically challenging. This should be elaborated more with reference to endotracheal intubation and any differences in challenges with the newer HFNC technologies.
33. Patients managed with HFNCs have been treated with iNO in clinics, especially COVID-19 patients in recent years (DOI: 10.1177%2F11795484211047065). Please elaborate further the benefits achieved by the current implementation.
44. Images A-F in Figure 1 have poor resolution. Higher resolution images would be preferable for journal publication.
Comments on the Quality of English LanguageThe manuscript would benefit from minor revision of the English grammar. Review by a proofreading software such as Grammarly would be sufficient to address these concerns.
Author Response
Cousin et al. report a novel non-invasive methodology for delivery of inhaled nitric oxide (iNO) through high-flow nasal cannulas (HFNCs) for potential therapeutic application in spontaneously breathing patients. iNO is of significant interest as a rescue measure for patients at risk of decompensation and endotracheal intubation. Critically, little work has described the combination of iNO with HFNCs. iNO administration has recently been described with HFNCs for ARDS in COVID-19 patients, but in the early stages of research conflicting results have often been found on the efficacy of the technology for improved clinical outcomes (DOI: 10.1177%2F11795484211047065). Cousin et al. report a novel device configuration strategy for achieving iNO therapy with configurations applicable to both adult and pediatric patients which may serve as a step forward in investigating the efficacy of the combination strategy in clinical outcomes.
This manuscript is of high quality and suitable for publication in Advances in Respiratory Medicine following consideration of some minor revisions:
- In the introduction, several examples of the benefits of iNO administration are listed for pulmonary and cardiovascular conditions. Authors should also tie in the recent interest in iNO for treatment of respiratory infections and further comment on the utility of the enclosed technology for such applications. Some prior work is described by: Biomedicines 2022Feb 3;10(2):369 and Applied Materials Today22(2021) 100887.
As suggested by the reviewer, we added to the manuscript the potential anti-microbial role of nitric oxide in addition to its role on the pulmonary vasculatur.
- iNO treatment is stated as technically challenging. This should be elaborated more with reference to endotracheal intubation and any differences in challenges with the newer HFNC technologies.
We briefly elaborated on the challenge of iNO delivery in the manuscript. The main challenges will be the absence of simple available description of non-invasive iNO administration settings in current pediatric litterature, especially for high-flow nasal cannulas.
- Patients managed with HFNCs have been treated with iNO in clinics, especially COVID-19 patients in recent years (DOI: 10.1177%2F11795484211047065). Please elaborate further the benefits achieved by the current implementation.
We discussed the recent study by Chandel et al. on the use of iNO trough HNFC in adult COVID-19 patients as you suggested. We think that an easier access to non-invasive iNO may benefit the development of further study using this agent, which may have multiple therapeuthic applications. The study by Chandel et al. that you mentioned clearly underlined the interest of clinicians for non-invasive application of iNO. Half of the patients positively respond to the iNO and it may suggest that judicious selection of patients may be crucial to observe a benefit from this therapy.
- Images A-F in Figure 1 have poor resolution. Higher resolution images would be preferable for journal publication.
As you suggested, we will discuss with the MDPI production team to know if they need new images.
Reviewer 2 Report
Comments and Suggestions for Authors
Overall, an interesting study dealing with an important aspect. In their manuscript, the authors presented a method to administer NO through HFNC. The authors provide useful technical information on how to assemble the circuit of the HFNC device to deliver NO; however, they did not collect any clinical information or show any data or comparison to test the effectiveness and feasibility of treatment administration.
They only briefly mentioned performing a bench test to test the "high flow nasal cannula configuration for iNO administration". I suggest better specifying information about bench testing (how it was performed?) and possibly collecting clinical data to show the effectiveness of this configuration on NO delivery.
Author Response
Overall, an interesting study dealing with an important aspect.
In their manuscript, the authors presented a method to administer NO through HFNC. The authors provide useful technical information on how to assemble the circuit of the HFNC device to deliver NO; however, they did not collect any clinical information or show any data or comparison to test the effectiveness and feasibility of treatment administration.
They only briefly mentioned performing a bench test to test the "high flow nasal cannula configuration for iNO administration". I suggest better specifying information about bench testing (how it was performed?) and possibly collecting clinical data to show the effectiveness of this configuration on NO delivery.
We test on a life-like dummy patient with HFNC in place the measured iNO. We agree with the reviewer that some clinical data are important, however our comments were mostly on the technical method to use iNO administration with commercially avaible HFNC kit. Clinical measures of such administration would have been outside the scope of this brief technical note. We are currently collecting data on patients treated with such device and will enventually publish them in a proper manuscirpt.
Round 2
Reviewer 2 Report
Comments and Suggestions for Authors
The manuscript has been improved. I would add a sentence at the end in the conclusionemphasizing the potential value of this system in clinical practice.
“Given the widespread use of HFNC in different clinical settings, including COVID-19 pneumonia and in complex patients with several comorbidities, these data might be of clinical importance for the scientific community.”
Related references:
doi:10.1183/13993003.01574-2021
doi:10.1007/s00134-020-06312-y
doi:10.1016/j.jcrc.2018.12.015.
doi: 10.1001/jama.2022.0028. PMID: 35072713; PMCID: PMC8787685.
doi: 10.1136/thoraxjnl-2022-218806. Epub 2022 May 17. PMID: 35580898.
Comments on the Quality of English Language
Fair
Author Response
We thank the reviewer for the suggestion.
We have insert a brief comment at the end of the conclusion as suggested to underline the role of NO administration in addition to HFNC support. We have added 2 references as we may have a limitation due to the nature of the manuscript, a Commentary. However, we agree that it is crucial to underline the expanding role of HFNC in clinical settings.